# The Lived Experiences of Fathers in Mealtimes: A Thematic Synthesis of Qualitative Literature

**DOI:** 10.3390/ijerph19021008

**Published:** 2022-01-17

**Authors:** Natalie Campbell, Michèle Verdonck, Libby Swanepoel, Laine Chilman

**Affiliations:** School of Health and Behavioural Sciences, University of the Sunshine Coast, Locked Bag 4 Maroochydore, Maroochydore DC, Sunshine Coast, QLD 4558, Australia; nataliegracecampbell@gmail.com (N.C.); michele.verdonck@usc.edu.au (M.V.); lswanepo@usc.edu.au (L.S.)

**Keywords:** fathers, lived experience, family meals, feeding behaviour, fussy eating, food fussiness, food refusal

## Abstract

The paternal experience of family mealtimes is an emerging field within qualitative literature. Previous quantitative studies suggest that differences exist between fathers’ and mothers’ mealtime behaviours, particularly in response to fussy eating. However, qualitative research has not yet focused exclusively on fathers’ fussy eating experiences. This metasynthesis aimed to provide insights into the general paternal experience, inclusive of their fussy eating responses. Thematic synthesis methodology was adopted to achieve this process and consisted of a systematic search resulting in the inclusion of 16 studies (18 papers). The direct quotations presented within each study were subjected to three stages of analysis to produce three analytical themes, supported by eight descriptive themes. The analytical themes presented were: (1) environmental influences on fathers’ mealtime experiences; (2) attitudes and emotions of fathers during mealtimes; and (3) observable behaviours of fathers during mealtimes. These themes highlighted the complexity of the mealtime experience from a paternal perspective. Multidirectional relationships were identified between each mealtime component (i.e., the environment, attitudes, emotions and behaviours) as evidenced by the paternal commentary presented. The findings also provided insights into fathers’ fussy eating experiences, recognising that fathers should be considered as individuals in the presence of mealtime intervention.

## 1. Introduction

Mealtimes are a universal experience characterised by the shared consumption of a meal by two or more participants. They consist of, and are influenced by, various intrinsic (internal) and extrinsic (external) factors which interact to produce either a successful or unsuccessful mealtime experience. Intrinsic factors pertain to the personal attributes of those involved, such as thoughts, motivations and behaviours, whereas extrinsic factors range from the interactions of participants to the state of their physical mealtime environment (i.e., location, presence of distractions). Considering the potential impacts of each factor on consumption, mealtimes are deemed a complex experience with the potential to either support or inhibit engagement in healthful feeding behaviour. 

Fussy eating is an intrinsic mealtime factor characterised by food rejection, limited food consumption and/or fluctuation in food preferences [1]. There remains a lack of consensus regarding the prevalence and trajectory of fussy eating, with researchers debating whether behaviours remain stable throughout childhood [2,3] or peak at a young age [4,5]. The impacts of fussy eating appear to be far-reaching, with studies documenting their influence on both food consumption [1,6,7,8,9,10,11,12,13] and family mealtime dynamics [1,14,15,16,17]. Children and adolescents who engage in fussy eating behaviours are suggested to have a lower intake of vegetables [7,9,10,12], proteins [9,12] and various micronutrients (predominantly zinc and iron) [7,9]. As recognised in a review by Taylor and Emmett [7], these dietary characteristics place individuals at a greater risk of poor growth outcomes. Additionally, associations have been identified between fussy eating behaviour and the presence of a negative emotional climate during mealtimes. Concern [1,14,15,17,18] and frustration [1,14,16,17] are the two most prevalent responses described by parents and have been identified as contributors to their feeding practices [1,16,18]. 

Both quantitative and qualitative literature have documented the feeding practices adopted by parents in response to food fussiness. Examples include food pressuring [1,14,17,19,20], the preparation of alternative meals [1,14,17,21], the use of food or non-food rewards [14,17,19,22] and the accommodation of children’s food preferences during meal preparation [1,17,23]. A qualitative study conducted by Wolstenholme et al. [17], suggested that these practices vary across families and are likely to change over time in response to their perceived effectiveness. Variance has also been recognised within family units, with mothers and fathers offering their own unique approaches to feeding behaviour. 

Paternal involvement is an emerging field within the fussy eating literature. Quantitative research suggests that fathers engage in more coercive feeding practices than their female counterparts [22,24]. According to Harris et al., these behaviours may arise from a pragmatic approach rather than a concern for child dietary intake, as evident within a maternal population [22]. Additional associations are apparent between fathers’ use of rewards and child fussy eating [25], as well as their use of coercive feeding when anxious [26]. 

Few qualitative studies have explored father’s mealtime experiences through a fussy eating lens. Instead, research has focused on fathers’ general mealtime experiences to gather an understanding of their engagement. To date, only two reviews of this literature base have been conducted [27,28]. These narrative reviews have both explored paternal contributions to the mealtime experience and their associations with the feeding behaviours of other mealtime participants. Similar to the findings previously reported, fathers were identified as having different mealtime approaches to their partners [27,28] and influenced their children’s food consumption [28]. Whilst these existing reviews provide a broad understanding of the paternal experience supported predominantly by quantitative literature, they do not delve deeply into fathers’ firsthand accounts. The systematic searching and analysis of qualitative literature would provide additional depth and offer fresh insights into paternal mealtime engagement, inclusive of their fussy eating responses. The current metasynthesis aims to achieve this deepened understanding in accordance with the following research question: ‘What are the lived experiences of fathers in mealtimes?’

## 2. Materials and Methods

Metasynthesis involves the summation and interpretation of findings from numerous qualitative studies to achieve new understandings of an experience or phenomenon [29]. This design was suitable for the current study as it enabled the development of a comprehensive understanding of fathers’ mealtime experiences. With the ability to inform health policy [30,31], this research design also supported the identification of potential avenues for allied health intervention within the realm of fussy eating. 

Thematic synthesis was selected as the methodology for this metasynthesis [30]. Thematic synthesis has been used extensively within the field of paediatric health to explore topics such as child behaviour [32], treatment adherence [33] and the lived experience of various clinical conditions [34,35,36]. This method is based on the philosophy that one’s perceptions shape their knowledge of reality and features elements of both meta-ethnography and grounded theory. 

### 2.1. Selection Criteria

This metasynthesis included original qualitative and mixed-methods research published in peer-reviewed journals between January 2010 and January 2021. To be considered for inclusion, each study was required to include the paternal experience of family mealtimes, with a particular focus on activities completed within the home environment (i.e., meal preparation and feeding). The perspectives of both fathers and their immediate families were included to promote a balanced understanding of paternal behaviour. Participants were required to have at least one child 18 years or under.

The exclusion criteria were as follows:Focus exclusively on activities beyond the family mealtime environment (i.e., food shopping, school meals, snacking and beverage consumption);Focus on the mother’s role in mealtimes or provide direct commentary on mother–child feeding interactions;Discuss mothers and fathers as a collective (i.e., no separation of maternal and paternal perspectives);Focus on underlying medical conditions (e.g., disordered eating, autism spectrum disorder, obesity or dysphagia), dietary intake and/or weight, health impacts of family meals or the evaluation of mealtime interventions;Non-English studies.

### 2.2. Search Strategy and Screening

A series of keywords were developed in accordance with the PICO framework and modified for use across three online databases (Scopus, CINAHL and PubMed). The final search terms are depicted within Table 1. A comprehensive search of the literature was conducted in December 2020 and resulted in a yield of 1753 studies, with an additional three articles gathered through hand searching. Following duplicate deletion, a total of 908 studies remained.

See Figure 1. Screening was conducted in three stages using Rayyan QCRI, software developed to conduct systematic reviews in teams [37]. Title screening was undertaken initially by N.C., and the results were reviewed by a second member of the research team (L.C.). Both reviewers (N.C. and L.C.) then conducted independent abstract screening. Conflicts were flagged and discussed until consensus was achieved. Any further discrepancies were resolved in consultation with a third reviewer (M.V.). The same methods used throughout abstract screening informed the screening of full-text articles. Following this process, the review identified 18 eligible papers documenting the experiences of fathers in family mealtimes. 

### 2.3. Quality Appraisal

Although metasynthesis and thematic synthesis methods do not require quality appraisal [29], the research team decided to include this step to maximise the quality of this review. The Critical Appraisal Skills Programme (CASP) Qualitative Studies Checklist was used to evaluate the methodological strengths and limitations of the included studies [38]. Quality appraisal was conducted by three independent reviewers (N.C., L.C. and L.S.). The research team discussed inconsistencies across the appraisals and reached consensus on the overarching quality of each study. 

In accordance with Noyes et al. [39], quantitative measures were avoided when determining the overall quality of each study. Through group discussion, the studies were instead allocated to one of three categories of perceived quality (‘low’, ‘moderate’ or ‘high’). 

### 2.4. Data Analysis

Data analysis was managed using NVivo 12. Only first order constructs (direct quotations) were analysed as the primary data within each paper. Some studies provided commentary on experiences beyond that of the father (i.e., the mealtimes of grandparents and children). Any direct quotations that were irrelevant to the paternal experience were not considered for analysis. Thematic synthesis encourages the preservation of data published within original qualitative literature whilst promoting the development of new interpretations [30]. Three key stages were adhered to throughout the synthesis process: (1) line-by-line coding of text from primary studies; (2) grouping of similar codes to develop descriptive themes; (3) development of analytical themes, guided by the research question. 

Line-by-line coding according to meaning and content was conducted independently by two reviewers (N.C. and L.C.). Following the coding of one paper, the reviewers met to discuss their codes and check for consistency in their coding techniques. After consolidation, the reviewers then continued with independent coding, meeting weekly to share newly established codes and come to an agreement regarding their inclusion. During this process, each reviewer also began to arrange the codes into a hierarchical structure. 

The development of descriptive themes involved regular collaboration between members of the research team. Following line-by-line coding, the team met to brainstorm similarities between the existing codes and arrange them into a basic hierarchy. Regular meetings were then held to support further development of the coding structure. 

In the final stage of analysis, the research team interrogated the descriptive themes and their associated quotes in relation to the research question to determine a series of analytical themes. 

## 3. Results

Sixteen studies conducted across North America [40,41,42,43,44,45,46,47,48,49], Europe [50,51,52,53] and Australia [54,55,56,57] were included in this metasynthesis. Sample sizes ranged from 8 participants [40] to 149 participants [47], with studies exploring the perspectives of fathers, mothers and families. Semi-structured interviews and focus groups were used for data collection and thematic analysis was a preferred method for data analysis. A summary of these characteristics is provided in Table 2.

The results of quality appraisal are depicted in Table 3. All studies featured clear research aims and findings, and the appropriate selection of qualitative methodology. Most studies identified and outlined the processes of recruitment and data collection in detail, demonstrating their relevance to the research question. Those areas of lower quality were the researcher–participant relationship and rigour of the analysis process. Only three studies described the researcher–participant relationship in adequate detail [41,44,53]. Insufficient detail was also present across five accounts of the analysis process [42,45,50,51,52]. Whilst ethical approval was described by the majority, two studies presented a lack of clarity in their reporting [50,52] and one did not provide an ethical statement [51]. In alignment with the methods presented by Thomas and Harden [30], the research team opted to include all studies despite the presence of varied quality.

The analysis of eighteen articles resulted in the identification of 36 initial codes describing fathers’ experiences of mealtimes. The initial codes were reviewed and arranged into eight descriptive themes. Regular discussion and interrogation of these themes in relation to the research question led to a consensus and the establishment of three analytical themes: (1) environmental influences on fathers’ mealtime experiences; (2) attitudes and emotions of fathers during mealtimes; and (3) observable behaviours of fathers during mealtimes. Each analytical theme was supported by four to five descriptive themes. Verbatim quotes that best represented each theme were presented both in text and within Table 4, Table 5 and Table 6.

### 3.1. Environmental Influences on Fathers’ Mealtime Experiences

The first analytical theme provided insights into the external influences on mealtime engagement from a paternal perspective. Three influences were identified and explored through the following descriptive themes: (1) family collaboration shapes the mealtime experience; (2) my past experiences influence mealtimes today; (3) time dictates how we spend mealtimes as a family. 

#### 3.1.1. Family Collaboration Shapes the Mealtime Experience

The first environmental influence was fathers’ collaboration with their families. Collaboration was defined across studies as the combined efforts of the family to complete meal-related tasks [40,42,46,49,51,53,54,55,56,57]. In the presence of disagreement, families who did not collaborate appeared to experience prolonged exposure to mealtime conflict. Two primary sources of disagreement amongst families were the presence of technology-based distractions at the dinner table and the refusal of new or unpreferred foods. Fathers perceived technology as a barrier to family connection during mealtimes [40,43,54,55] and shared their attempts to regulate its presence at the dinner table [40,43,54]. Children frequently responded to these attempts with resistance, demonstrated by a continuation of device use [40] or the initiation of an argument [54]. For these fathers, the physical environment (technology) was perceived as a source of poor collaboration, impacting their ability to achieve positive mealtime interactions and outcomes. 

Food refusal was another source of conflict which arose when fathers rejected mothers’ attempts at introducing healthier meals. Some mothers described their partners as being resistant to their attempts at improving dietary quality [41,44] and recognised the influence of this lack of collaboration on their avoidance of implementing future food changes.

In contrast, families who regularly engaged in collaborative behaviour made no mention of conflict and instead commented on the contributions of collaboration to mealtime success. Some fathers described how collaboration began prior to the evening meal and consisted of joint discussions to identify the foods best suited to their family’s dietary needs and preferences [41,54,57]. This collaboration demonstrated a proactive approach to mealtimes, with parents attempting to maximise consumption and minimise food-based conflict. For example, one father said, “*We make sure we have good communication … to see what he wants to eat … we try to make a good variety of food [available] as well*” [54] (p. 4). Fathers also described task division and the use of shared feeding approaches as examples of effective family collaboration. Meal-related tasks (particularly meal preparation) were shared between couples in an attempt to reduce the burden of domestic duties on the primary caregiver [40,42,46,51,53,55,57]. Some fathers also recognised that in adopting a shared feeding approach with their partner, they were able to collectively identify and use strategies to respond to the challenges posed by food fussiness [46,48,53,54,57].

#### 3.1.2. My Past Experiences Influence Mealtimes Today

Fathers’ experiences of mealtimes during childhood were also identified as an environmental influence on their current mealtime structure and feeding practices. Those who described a continuation of mealtime traditions into adulthood reflected positively on their upbringing and demonstrated a value for the practices used by their parents. For example, one father stated, “*[Vegetables were] a strong point in my family so we have a balance where I … make sure that [the children try] to eat something green from time to time.*” [46] (p. 1065).

For some, traditions were characterised by mealtime formality and structure, with fathers requesting that their family eat meals together at the dinner table [45,54,55,57], whereas others expressed the adoption of their parents’ feeding practices [46,52,57] and their preference for consuming familiar foods [40,41,44,56]. 

In contrast, some fathers reported a lack of paternal involvement in their mealtimes as children [46,52]. Negative perceptions of this involvement prompted fathers to redefine the mealtime experience. Fathers chose to become more involved in family mealtimes by adopting new roles within the kitchen [46,52] and taking interest in their children’s feeding behaviours [48,52]. 

#### 3.1.3. Time Dictates How We Spend Mealtimes as a Family

Time was the third environmental descriptive theme. Busy family schedules posed time constraints on mealtime engagement during the working week. Fathers identified that their ability to coordinate work and school-based schedules determined whether they shared evening meals with their families. Clashes between family members’ schedules often lead to significant difficulties in mealtime coordination. Some fathers stated that they had resorted to eating alone [50,55]. Other fathers, however, developed strategies that enabled them to accommodate busy schedules and maintain family mealtimes [40,49,50,53,55]. 

Fathers identified a preference for weekend meals as they provided an opportunity for fathers to relax and enjoy mealtimes with their families outside of the constraints posed by full-time work [50,55]. One father shared, *“…it’s much more different [on weekends] … feeling not rushed for time, we don’t have to go to school or work”* [55] (p. 50).

### 3.2. Attitudes and Emotions of Fathers during Mealtimes

The second analytical theme encompassed the attitudes and emotions of fathers during mealtimes, which were supported by the following two descriptive themes: (1) mealtimes are an emotionally rich (and sometimes challenging) experience; and (2) my attitude informs my mealtime experience.

#### 3.2.1. Mealtimes Are an Emotionally Rich (and Sometimes Challenging) Experience

Mealtimes were an emotional experience shaped by fathers’ perceptions of family routine and behaviour. Many fathers identified mealtimes as a positive emotional experience and had a preference for sitting with and talking to their families during evening meals [42,45,49,54,55,57]. These occasions were recognised as a source of enjoyment as they were one of few opportunities for valued family interaction in the presence of busy weekday schedules [42,45,54,55,57]. 

In contrast, negative emotions such as frustration and worry were also reported [40,51,54,55]. These emotions were not described as apparent from the outset of the meal, but rather arose in response to disruptive child behaviours [40,51,54,55]. Food refusal was a primary source of frustration as fathers felt their cooking efforts had been disregarded by their children [40,51,54]. For some, this frustration translated to worry as they considered the long-term effects of refusal on their child’s dietary intake [50,52,54,57]. Whilst influenced by child behaviour, worry also guided fathers’ feeding practices and resulted in pressuring [47,50] and catering to children’s food preferences [52,54]. Some fathers were also concerned about how to best support their child’s dietary intake, stating *“…are there any foods that are a ‘must have’? … he eats so variably from day to day and week to week.”* [57] (p. 8).

#### 3.2.2. My Attitude Informs My Mealtime Experience 

Fathers’ attitudes informed their reported mealtime responses. Health consciousness and pragmatic attitudes were regularly adopted during mealtimes and informed fathers’ dietary choices and feeding behaviours. Those who demonstrated an awareness for health and nutritional intake commented on their inclusion of vegetables in the evening meal [40,44,46,51,57] and identified the importance of engaging their children in meal preparation [44,51]. Fathers also described their monitoring of food intakes, with one participant stating, *“I’m really conscious of the amount of food me and my son eat…”* [43] (p. 143).

Pragmatic attitudes were described as a desire to complete the mealtime process in a timely manner with minimal hassle. In order to achieve this outcome, fathers opted for the use of premeditated plans or the introduction of undesired tactics. Fathers who favoured premeditated plans commented on their desire for their children to eat everything on their plate [40,47,50,54]. When faced with refusal, these fathers were firm in their approach and did not provide a meal alternative [40,54]. Conversely, refusal led other fathers to adopt undesired feeding tactics such as providing their child with a desired meal of lower nutritional quality [40,51,52,54].

### 3.3. Observable Behaviours of Fathers during Mealtimes

The third analytical theme explored a variety of paternal behaviours associated with mealtimes. These behaviours were all described in relation to child behaviour, with a focus on consumption and nutritional education. The three descriptive themes include: (1) behaviours to make sure that my children eat; (2) mealtimes are an opportunity for teaching and exploration; and (3) I use set strategies in response to my child’s food refusal.

#### 3.3.1. Behaviours to Make Sure That My Children Eat

Fathers described three primary behaviours which they used to encourage their children’s food consumption. These behaviours involved allowing children to make their own food choices [43,46,49,52,54], providing alternative meals [52,54] and using rewards as external motivation [43,49,52,57]. Those who encouraged their children to make independent food choices did so as an alternative to pressuring consumption [43,46,52,54]. These fathers recognised food preference as a determinant of their children’s willingness to eat [43,46,49,52,54], and chose to accept food rejection when presenting undesired foods [43,46,52,54].

Fathers provided alternative meals as a last resort to encourage consumption amongst children who were frequently resistant to eating the evening meal [52,54]. These behaviours included the provision of foods with a lower nutritional content. One father stated, *“My daughter, sometimes she won’t eat anything, and if she wants someat (something) I grab at that chance, because I have always made sure she eats.”* [52] (p. 403).

Most fathers described how they used sweet foods as an incentive for their children to finish the evening meal [43,49,52,57]. Thus, using food rewards as external motivation. These behaviours were adopted in response to mealtime resistance, with fathers reinforcing the rule that dessert would not be allowed until the meal was finished [43,52,57]. 

#### 3.3.2. Mealtimes Are an Opportunity for Teaching and Exploration 

Modelling and face-to-face discussions were used to educate children on food consumption and appropriate mealtime behaviour. Fathers demonstrated an awareness of their behavioural influence on their children’s development and recognised the importance of modelling healthy and mature practices during shared meals [42,44,51,54,55,57]. Vegetable consumption was of particular interest, with fathers noting that they regularly modelled this behaviour to encourage their children to eat healthy, balanced meals [44,57]. Fathers also expressed the need to model formalities such as sitting at the dinner table, in an effort to teach their children mealtime etiquette. One father stated, *“We make sure that we don’t leave the table at dinnertime … until everyone’s finished, so introducing … politeness and things.”* [57] (p. 6).

In comparison to modelling, face-to-face discussions focused on food and its nutritional quality and associated health benefits. Some fathers chose to discuss food in an interactive setting, using meal preparation as an opportunity to guide their children’s exploration of new foods [51], whereas others focused on discussing food at the dinner table, responding to queries as they arose [43,49,57]. In this instance, discussions were centred around dietary intake and explored the reasoning behind mealtime decisions. 

#### 3.3.3. I Use Set Strategies in Response to My Child’s Food Refusal

Food refusal was described as a frequent barrier to mealtimes, with fathers utilising a variety of strategies in response to resistance. Of the responses presented, fathers were found to either alter their mealtime practices or modify the feeding environment. 

Altering mealtime practices included the acknowledgement of food preferences [52,54], use of pressuring [43,47,49,54] and introduction of compromise [47]. These responses reflected practical approaches to mealtimes, with fathers attempting to complete the meal in the method deemed most appropriate to their circumstance. In some cases, the success of these methods was questioned, with fathers demonstrating the desire for better alternatives [52,54,57]. For example, *“…I suppose I’m looking for her to eat a bit more … I don’t like doing it but I’ve got no other tools.”* [57] (p. 11).

With regard to environmental modification, some fathers attempted to make food more appealing by using creativity [49,51] or hiding undesired foods amongst children’s preferences [43,54], whereas others commented on the consistent provision of desired food in an attempt to avoid mealtime conflict and support consumption [51,52,54]. Several fathers showed a desperation for their children to eat and as a result, focused on mealtime quantity over quality. This behaviour was apparent when one father stated, *“…if they don’t eat or if they are not eating well, you start cutting corners and getting takeaways…”* [54] (p. 5).

## 4. Discussion

The aim of this metasynthesis was to provide a broadened understanding of the lived experiences of fathers in mealtimes. Through thematic synthesis, eighteen papers were collected and analysed to produce descriptive and analytical themes. Three analytical themes illustrated the thoughts, behaviours and environmental influences present during mealtimes and their influences on paternal involvement. Supported by eight descriptive themes, these findings provide an overarching understanding of the mealtime experience, highlighting its complexity and the unique contributions offered by fathers. 

The findings presented in this metasynthesis demonstrate how the interactions of fathers’ thoughts, behaviours and environments can influence their experiences of family meals. Fathers commented on the past and present influences of family interaction on their current mealtime engagement. The influence of past experience has been previously addressed within the literature, with parents describing their continuation of mealtime traditions from childhood [58]. Within this review, fathers’ descriptions of the past were emotional in nature as they commented on how their negative perceptions of childhood meals prompted change.

An emotional component was also evident within fathers’ descriptions of their current mealtime interactions. Fathers’ emotional responses were linked to technology [40,54,55], which was recognised as a barrier to family connection. This finding is supported by Nelson [59] who identified that technology use during meals affects fathers’ feelings of closeness with their families. Given that fathers described mealtime enjoyment as the opportunity to connect with other family members, it is possible that technology-based conflict had a multidirectional influence on both mealtime engagement and paternal emotion. Time constraints were described as having a similar impact. Fathers who chose to eat alone [50,55] did not have the opportunity to achieve the same family connections and enjoyment as those who were able to accommodate their family’s busy schedules [40,50,53,55]. It is important to note that these interpretations do not imply a correlational component within the current findings. Rather, they encourage a consideration of the multidirectional relationships present during family meals. 

The term ‘multidirectional’ is used in this context to describe the numerous influences presented by each factor of the mealtime experience (i.e., observable behaviours, environmental influences, attitudes and emotions). In comparison to existing research, which has focused on the bidirectionality of parent–child feeding interactions [20,60,61,62,63], this metasynthesis encourages one to adopt a broader perspective. Each component of the mealtime experience has the potential to influence numerous others. For example, fathers’ attitudes impact and are impacted by environmental influences, which in turn can produce a series of observable behaviours and elicit an emotional response (see Figure 2).

The findings also highlight fathers’ experiences of fussy eating, offering insights into how perceptions of child behaviour might affect fathers’ emotional responses and observable behaviours. The negative emotions presented within this review were both described in relation to children’s food refusal. As supported by the literature, these are common responses to fussy eating behaviour for both mothers and fathers [1,16,18,22,28]. Whilst fathers may express a lower level of worry about their children’s dietary intake than mothers [22], the current findings highlight that paternal concern is still prevalent. As suggested by Rahill et al. [28], these concerns are associated with fathers’ perceived responsibility for feeding, which may explain why the majority of behaviours described by fathers were adopted in response to fussy eating. 

The cooking of alternative meals [1,21], use of food rewards [22,24,64], and pressuring [18,22,24] have been explored within the existing literature as common behavioural responses to fussy eating for fathers. Commentary on these behaviours within this review highlighted that there is not a ‘one size fits all’ approach to fussy eating behaviour. Differences remain present amongst paternal populations with some fathers reporting that they use a proactive approach (i.e., hiding undesired food) whilst others respond when behaviours arise. This outlook shows the importance of considering personal experience when providing mealtime-based intervention. 

### 4.1. Limitations

Several limitations were present in the current metasynthesis. Fathers recruited in the original qualitative studies may have been more ‘involved’ than the average paternal population, introducing the potential for participant bias [65]. Whilst mothers’ perceptions were included in this review (as evidenced in the search terms), those that focused exclusively on mother–child feeding interactions were excluded. “Mother*” and “maternal” search terms were adopted to obtain studies on mothers’ perceptions, but only in the broader context of family meals (inclusive of their partners’ behaviours). Additionally, whilst two of the studies explored maternal perceptions of fathers’ behaviour, the majority focused on paternal perceptions in isolation. Therefore, it is recognised that the findings presented in this review may be biased towards paternal perceptions and may not provide a true reflection of paternal behaviour as perceived by other family members. Non-English studies were excluded from this review as access to adequate translation services during the research process was not available. Because of this, the body of literature available for inclusion was reduced and may have featured a predominance of Westernised perspectives. The inclusion of all studies regardless of quality impedes upon the reliability of the findings presented within this review. Finally, given that metasyntheses rely solely on the use of secondary data that have been previously analysed, the results presented reflect the interests of past researchers and may not be a complete reflection of the paternal experience.

Three members of the research team (L.C., M.V., L.S.) were parents to young children and had their own perceptions of fathers’ mealtime behaviours. The other researcher (N.C.), whilst not a parent, had attained firsthand experiences of fussy eating and paternal behaviour throughout her childhood, with family meals featuring sibling fussiness and coercive parenting. The research team adhered to the four criteria for trustworthiness—credibility, confirmability, dependability and transferability—to minimise any negative outcomes posed by these influences [66,67].

### 4.2. Clinical Implications and Recommendations

The findings presented by this metasynthesis provide insights into the perceived complexity of family mealtimes for both fathers and their immediate families. In considering the multidirectional nature of mealtimes, clinicians are encouraged to adopt a holistic approach to intervention. As fathers have also been recognised for their unique contributions to both general mealtime practices and fussy eating behaviour, it is recommended that clinicians consider family-based therapy in their response to issues such as mealtime cohesion and fussy eating. 

As it is an emerging area of study, future research should explore the multidirectional nature of mealtimes further. Whilst this metasynthesis poses the presence of multidirectional relationships within fathers’ mealtime experiences, it is currently unknown as to whether these relationships are present for mothers. If such relationships are present, it would also be beneficial to explore how they influence the maternal experience to identify whether mothers’ and fathers’ experiences differ. The maternal perception of fathers’ mealtime practices is also an emerging field, with two studies in this review commenting on the perceptions of Hispanic mothers [41,44]. To promote a balanced understanding of paternal behaviour, it is recommended that future research explores maternal perceptions from a variety of ethnic and socioeconomic backgrounds.

## 5. Conclusions

Through thematic synthesis, this review explored fathers’ lived experiences of family mealtimes. The results highlighted the influences of various mealtime factors on the paternal experience, including the environment, fathers’ attitudes and emotions, and their observable behaviours. Exploration of these mealtime components in isolation aided the development of a new interpretation—the interactions of mealtime components are multidirectional in nature. Additionally, fathers were recognised for their unique mealtime contributions. In considering both of these findings, health professionals are encouraged to consider holistic, family-based interventions in response to mealtime challenges for both fussy and non-fussy eaters. 

## Figures and Tables

**Figure 1 ijerph-19-01008-f001:**
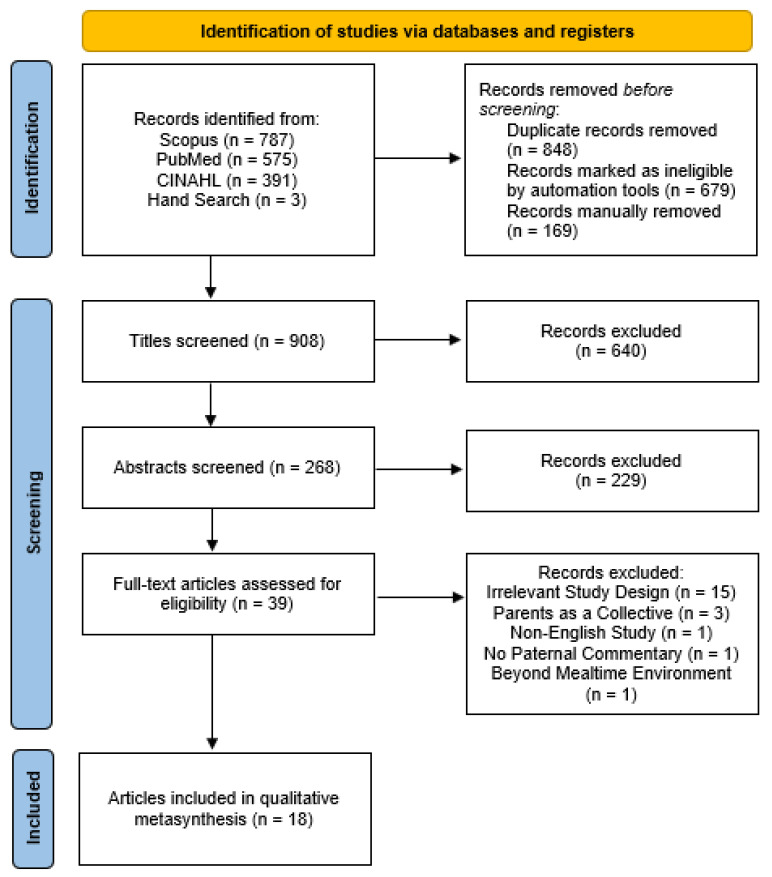
PRISMA diagram.

**Figure 2 ijerph-19-01008-f002:**
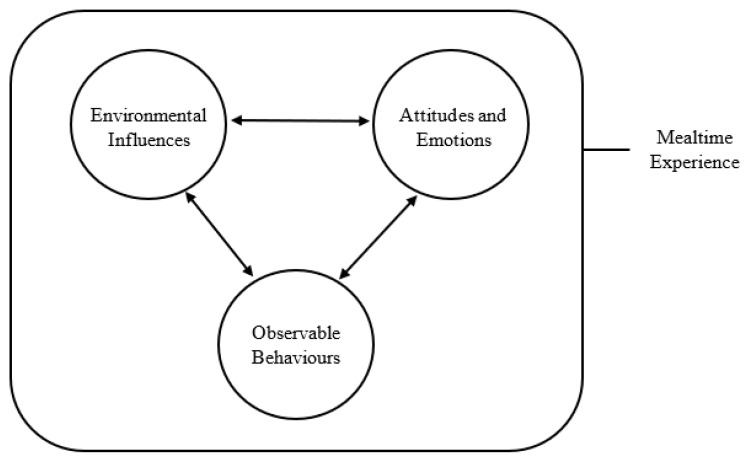
The multidirectional nature of mealtimes.

**Table 1 ijerph-19-01008-t001:** Search terms aligned with PICO framework.

Population	Interest	Context
“father” OR “paternal” OR “mother” OR “maternal” OR “parent *” OR “coparent *”	meal *” OR “breakfast*” OR “lunch *” OR “dinner*” OR “supper *” OR “meal prep*”	“role *” OR “involvement” OR “participation” OR “responsibilit *” OR “food practices”

*: Star indicates truncation, which was used as a technique to broaden the search to include various word endings.

**Table 2 ijerph-19-01008-t002:** Characteristics of selected literature.

Study	Location	Demographic Information	Sampling Method	Data Collection	Analysis	Study Focus
Owen et al., 2010 [52]	UK	Fathers (n = 29) with children aged 5–11, living in contrasting socio-economic areas	Not made explicit	Semi-structured interviews, observations and photos	Not made explicit	Fathers’ and children’s perspectives on food practices
Brannen, O’Connell and Mooney, 2013 [50]	UK	Dual-earner households (n = 40) with children aged 18 months–10 years	Recruited from another study	Semi-structured interviews	Not made explicit	The synchronisation of family schedules in relation to weekday mealtimes
Del Bucchia and Peñaloza, 2016 [51]	Switzerland	Parents (n = 21; 13 mothers, 8 fathers) in charge of meal preparation	Purposive and snowball	Semi-structured interviews using photo elicitation	Cross-case analysis	Parents’ understandings of themselves and their practices in the context of family meals
Khandpur, Charles and Davison, 2016 [42] *	USA	Fathers (n = 37) of children aged 2–10 years	Purposive and snowball	Semi-structured interviews	Thematic analysis	Fathers’ perceptions of food parenting tasks completed by themselves and their partners
Khandpur et al., 2016 [43] *	USA	Fathers (n = 40) of children aged 2–10 years	Purposive and snowball	Semi-structured interviews	Thematic analysis	The food parenting practices used by a heterogeneous sample of fathers
Rhodes et al., 2016 [56]	Australia	Three generation families (n = 27), with Anglo-Australian (n = 11), Chinese-Australian (n = 8) or Italian-Australian (n = 8) heritage	Purposive	Semi-structured family interviews	Thematic analysis	Food-related decision-making and behaviour within a broad family context
Thompson et al., 2016 [53]	UK	Parents (n = 9; 8 mothers, 1 father) with young and/or school-aged children from a low socio-economic area	Not made explicit	Semi-structured interviews using photo elicitation	Thematic analysis	Parents’ thoughts and responses to child food preferences during family meals
Lora, Cheney and Branscum, 2017 [44]	USA	Hispanic mothers (n = 55) of children aged 2–5 years	Purposive	Nine focus groups	Grounded theory and thematic analysis	Hispanic mothers’ views on paternal health promotion at home
Walsh et al., 2017 [57]	Australia	Fathers (n = 20) of children 5 years and under from diverse socio-economic backgrounds	Purposive stratified and snowball	Semi-structured interviews	Thematic analysis	Fathers’ perceptions and involvement in their children’s eating and physical activity behaviours
Zhang et al., 2018 [49]	USA	Latino fathers (n = 26) with children aged 1–14 years	Convenience	Four focus groups	Thematic analysis	Perspectives and practices of Latino fathers regarding their teens’ eating, physical activity and screen time behaviours
Greder et al., 2020 [40]	USA	First-generation Mexican immigrant fathers (n = 8) with a child aged 6–18 years	Not made explicit	Two focus groups	Thematic analysis	Mexican fathers’ perceptions, behaviours and roles in relation to family mealtime consumption and physical activity
Harris, Jansen and Rossi, 2020 [54] ^#^	Australia	Fathers (n = 27) with children 12 and under, employed in service industries or blue-collar occupations	Convenience	Six focus groups	Grounded theory	Fathers’ lived experiences of family mealtime interactions
Jansen, Harris and Rossi, 2020 [55] ^#^	Australia	Fathers (n = 27) with children 12 and under, employed in service industries or blue-collar occupations	Convenience	Six focus groups	Grounded theory	Fathers’ negotiation of feeding roles and their impact on mealtime structure
Méndez et al., 2020 [45]	USA	Hispanic and non-Hispanic parents (n = 32; 29 mothers, 3 fathers) of primary students	Not made explicit	Four focus groups	Thematic analysis	Mexican and non-Hispanic parents’ perceptions of mindful eating and food parenting
Tan et al., 2020 [46]	USA	Heterosexual couples (n = 30), married or cohabiting, with children aged 3–5 years	Convenience	Semi-structured interviews	Constant comparative method	Parents’ joint navigation of child feeding and their associated agreements/disagreements
Hammons et al., 2021 [41]	USA	Mexican and Puerto Rican mothers (n = 46) with children aged 6–18 years	Not made explicit	Eleven focus groups	Thematic analysis	Mexican and Puerto Rican mothers’ perspectives on establishing healthy family meals
Trofholz et al., 2021 [47]	USA	Families (n = 149; 127 food secure and 27 food insecure) with a child aged 5–7 years	Not made explicit	Semi-structured interviews	Deductive and inductive content analysis	Meal characteristics and feeding practices of racially and ethnically diverse families
Walton et al., 2021 [48]	Canada	Dual-headed families (n = 20) with a child aged 18 months–5 years	Maximum variation	Semi-structured interviews	Thematic analysis	Influences of parents’ childhood eating practices on their current mealtime experiences and dynamics

^#^ Both papers from the same study. * Both papers from the same study.

**Table 3 ijerph-19-01008-t003:** Results of quality appraisal.

Study	1	2	3	4	5	6	7	8	9	Quality
Owen et al., 2010 [52]	Y	Y	Y	Y	Y	N	CT	N	Y	Low
Brannen, O’Connell and Mooney, 2013 [50]	Y	Y	Y	Y	Y	N	CT	N	Y	Low
Del Bucchia and Peñaloza, 2016 [51]	Y	Y	Y	Y	Y	N	N	N	Y	Low
Khandpur, Charles and Davison, 2016 [42]	Y	Y	CT	Y	N	N	Y	N	Y	Low
Khandpur et al., 2016 [43]	Y	Y	Y	Y	Y	N	Y	Y	Y	High
Rhodes et al., 2016 [56]	Y	Y	Y	Y	Y	N	Y	Y	Y	High
Thompson et al., 2016 [53]	Y	Y	Y	Y	Y	Y	Y	Y	Y	High
Lora, Cheney and Branscum, 2017 [44]	Y	Y	Y	Y	Y	Y	Y	Y	Y	High
Walsh et al., 2017 [57]	Y	Y	Y	Y	Y	N	Y	Y	Y	High
Zhang et al., 2018 [49]	Y	Y	Y	Y	Y	CT	Y	Y	Y	High
Greder et al., 2020 [40]	Y	Y	CT	Y	Y	N	Y	Y	Y	Moderate
Harris, Jansen and Rossi, 2020 [54]	Y	Y	Y	Y	Y	CT	Y	Y	Y	High
Jansen, Harris and Rossi, 2020 [55]	Y	Y	Y	Y	Y	N	Y	Y	Y	High
Méndez et al., 2020 [45]	Y	Y	Y	CT	Y	N	Y	N	Y	Low
Tan et al., 2020 [46]	Y	Y	Y	Y	Y	CT	Y	Y	Y	High
Hammons et al., 2021 [41]	Y	Y	CT	Y	Y	Y	Y	Y	Y	High
Trofholz et al., 2021 [47]	Y	Y	Y	Y	CT	N	Y	Y	Y	Moderate
Walton et al., 2021 [48]	Y	Y	Y	Y	Y	CT	Y	Y	Y	High

CASP checklist questions were as follows: (1) Was there a clear statement of the aims of the research? (2) Is a qualitative methodology appropriate? (3) Was the research design appropriate to address the aims of the research? (4) Was the recruitment strategy appropriate to the aims of the research? (5) Were the data collected in a way that addressed the research issue? (6) Has the relationship between researcher and participant been adequately considered? (7) Have ethical issues been taken into consideration? (8) Was the data analysis sufficiently rigorous? (9) Is there a clear statement of findings?. Abbreviations: Y = Yes; CT = Can’t Tell; N = No.

**Table 4 ijerph-19-01008-t004:** Representative quotes for first analytical theme: environmental influences on fathers’ mealtime experiences.

Descriptive Theme	Representative Quote	Reference
Family collaboration shapes the mealtime experience	“… it was more teamwork … to create healthier eating habits.”	Zhang et al. [49]
	“One day, I said to him, if you don’t stop that iPad, I’m going to break that in front of you … if you’re around the table, you’re supposed to face each other, talk to each other.”	Harris, Jansen and Rossi [54]
My past experiences influence how I run mealtimes today	“… I just didn’t see the man cooking. And then as I’ve gotten older and I’ve started to cook, I actually enjoy it and I don’t mind doing it at all.”	Tan et al. [46]
Time dictates how we spend mealtimes together	“...time is the biggest thing you need to manage.”	Jansen, Harris and Rossi [55]

**Table 5 ijerph-19-01008-t005:** Representative quotes for second analytical theme: attitudes and emotions of fathers during mealtimes.

Descriptive Theme	Representative Quote	Reference
Mealtimes are an emotionally rich (and sometimes challenging) experience	“One likes to be with one another … it [the evening meal] is … the time to concentrate, to talk about what happened during the day.”	Méndez et al. [45]
	“Now I understand how frustrating it [fussiness] is … I spend an hour cooking something … put it down and they just look at it and turn their noses up.”	Harris, Jansen and Rossi [54]
	“…how do we know we’re getting it right?”	Walsh et al. [57]
My attitude informs my mealtime experience	“We don’t make them [the children] anything other than what is at home and since they are hungry, they eat it.”	Greder et al. [40]

**Table 6 ijerph-19-01008-t006:** Representative quotes for third analytical theme: observable behaviours of fathers during mealtimes.

Descriptive Theme	Representative Quote	Reference
Behaviours to make sure that my children eat	“We dangle the carrot of … a sweet or treat of some kind … to encourage the children to … eat the right foods”	Walsh et al. [57]
	“I’ll say to my son, you’re gonna give me a hard time over vegetables today, so pick a vegetable you’ll eat … and we’ll go home and cook it.”	Khandpur et al. [43]
Mealtimes are an opportunity forteaching and exploration	“Sweet potatoes, peanuts, fish, whatever her daddy puts in his mouth, [my daughter] puts in her mouth.”	Khandpur et al. [42]
	“It’s important that they discover new flavours, that they see the food before and after, and that we talk about it”	Del Bucchia and Peñaloza [51]
I use set strategies in response to my child’s food refusal	“You either force them to eat something, and they’ll rebel against it, or you hope that eventually they’ll try it”	Owen et al. [52]

## Data Availability

Data is available in associated peer reviewed publications as per Table 2.

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
