# Peer review of "The Lived Experiences of Fathers in Mealtimes: A Thematic Synthesis of Qualitative Literature"

_ijerph, 2022, doi:10.3390/ijerph19021008_

Round 1

Reviewer 1 Report

The paper is concisely written and talks about the important topic of the role of fathers’ fussy eating experiences. This aspect is especially relevant considering that most studies, both quantitative and qualitative, tend to focus on the mother. Therefore, it seems to be an article of great interest to better understand the dynamics that are established around mealtimes and the consequences that these can have for the development of obesogenic behaviours and as a cause of family conflicts.

Author Response

Thank you for your feedback. We greatly appreciate your comments on the relevance of our review and hope that our findings will continue to further understandings of the mealtime experience.

Reviewer 2 Report

The manuscript entitled “The lived experiences of fathers in mealtimes: A thematic synthesis of qualitative literature” deals with an interesting and timely topic since the role of fathers in child rearing has changed in recent years due to the increase in maternal employment. Specifically, this review focuses on understanding fathers’ mealtime experience and highlights its complexity.

The strong points of this manuscript are its layout and the clarity of presented contents. The review protocol and article selection process are well prepared and executed. Results are properly presented in tables and easy to understand. 

Just a minor comment.

It is not clear to me why the authors included the quality appraisal method but, in the end, they presented all papers in the review. As far as I know the quality appraisal is used in order to exclude studies of perceived low quality from the review or to classify studies in a hierarchical order and present results accordingly. Is it possible that classifying the articles by publishing year, would show how the literature has evolved in terms or research quality?

Author Response

Thank you for your feedback. We conducted quality appraisal to determine the quality of each included study, rather than using it as a method for exclusion. In doing this, we aimed to maximise transparency between ourselves and the reader.

As guided by your comments, we agreed that it would be beneficial to classify the articles by publishing year and have done so in our revised manuscript (see Tables 2 and 3). Interestingly, the table now demonstrates the evolution of research quality, with the majority of our ‘low quality’ studies being published between 2010 and 2016.